# Comparisons of Nutrient Intakes and Diet Quality among Water-Based Beverage Consumers

**DOI:** 10.3390/nu11020314

**Published:** 2019-02-01

**Authors:** Leila M. Barraj, Xiaoyu Bi, Mary M. Murphy, Carolyn G. Scrafford, Nga L. Tran

**Affiliations:** Exponent, Inc., Center for Chemical Regulation and Food Safety, Washington, DC 20036, USA; lbarraj@exponent.com (L.M.B.); xbi@exponent.com (X.B.), mmurphy@exponent.com (M.M.M.); ntran@exponent.com (N.L.T.)

**Keywords:** sugar-sweetened beverages, low- and no-calorie sweetened beverages, zero-calorie unsweetened beverages, diet quality, added sugar

## Abstract

Americans are encouraged to reduce intake of sugar-sweetened beverages (SSB). Zero and low-calorie water-based beverages can provide alternative options to SSB, though limited data are available to understand measures of diet quality across different beverage consumer groups. The purpose of this cross-sectional study was to quantify intake of added sugars, total sugars, carbohydrates, and diet quality among consumers of zero-calorie unsweetened beverages (ZCUB) compared to SSB; and, separately, among consumers of low- and no-calorie sweetened beverages (LNCSB) when compared to SSB. Dietary data from the 2009–2016 National Health and Nutrition Examination Survey (NHANES) were analyzed among three life stages by SSB, ZCUB, and LNCSB consumer groups and adjusted for participant characteristics and energy intake. Across all life stages, ZCUB and LNCSB consumers had lower mean intakes of total sugars, added sugars, and carbohydrates when compared to SSB consumers. Diet quality as measured by the Healthy Eating Index 2015 (HEI-2015) was also higher among ZCUB and LNCSB consumers compared to SSB consumers in analyses adjusted for participant characteristics. These results indicate that reduction of SSB from dietary patterns and replacement with ZCUB or LNCSB could help Americans improve overall diet quality.

## 1. Introduction

Sugar-sweetened beverages (SSB) are a source of both energy and added sugars in the diets of children and adults in the United States (US) [1,2,3,4]. Dietary data from the National Health and Nutrition Examination Surveys (NHANES) 2011–2014 indicate that approximately one-half of all adults and two-thirds of children in the US consumed at least one SSB on a given day, with these beverages accounting for an average of 145 and 143 calories per day for adults and children, respectively [3,4].

To help achieve healthy eating patterns, the 2015–2020 Dietary Guidelines for Americans (DGA) recommend reducing consumption of added sugars to less than 10% of calories per day. The DGA specifically advise Americans to reduce consumption of SSB and instead select beverages with low or no added sugars such as water [5]. The guidelines also acknowledge that coffee or tea with no added sweeteners or creamers are a minimal source of calories, and moderate consumption of caffeinated beverages can be a part of healthy eating patterns [5]. For individuals who do enjoy coffee and tea, water as well as plain coffee and tea are all zero-calorie (or essentially calorie-free) unsweetened beverage (ZCUB) options supported by dietary guidance.

Beverages sweetened with low- or no-calorie sweeteners (LNCSB) could provide yet another water-based beverage option with few or no calories and no added sugars. As noted in the DGA, individuals select products prepared with sugar alternatives such as low- and no-calorie sweeteners for various reasons, including a reduction in sugar intake [5]. Since release of the current DGA, the American Heart Association concluded in a Science Advisory that LNCSB may provide a helpful alternative for adults seeking to reduce consumption of SSB or for children managing diabetes [6]. Additionally, the Ibero-American consensus on low- and no-calorie sweeteners conclude, “It is proposed that foods and beverages with LNCS (low and no calorie sweetener) could be included in dietary guidelines as alternative options to products sweetened with free sugars” [7].

Current evidence from the US indicates that consumers of water or LNCSB do in fact have lower intakes of sugars and higher measures of diet quality in some [8,9] though not all [10] studies. In a cross-sectional study of children, diet quality as measured by the Healthy Eating Index 2010 (HEI-2010) was positively associated with water intake [8]. Using data from NHANES 2001–2012, increased consumption of water among adults was associated with lower total and added sugar intakes [9], while another study found that each percentage point increase in the proportion of daily plain water of total dietary water consumed was associated with lower sugar intakes but no change in diet quality [10].

Cross-sectional data in the U.S. have largely shown HEI scores for the SoFAAS component (i.e., translating to lower caloric intake from Solid Fats, Alcoholic beverages, and Added Sugars) [1]. Likewise, an analysis of beverage intake among adults in the UK indicated improved diet quality among consumers of LNCSB compared to consumers of SSB [11]. In contrast, consumers of any type of sweetened beverage, including LNCSB only consumers, had a lower probability of adhering to a prudent diet compared to non/low consumers of sweetened beverages in an analysis of dietary data from NHANES 2003–2010 [11,12]. Available data on beverage consumption patterns in the US suggest that patterns including water or LNCSB are associated with beneficial trends in nutrient intakes. However, data are lacking on comparisons between consumers of water or LNCSB and consumers of SSB to allow for a clear understanding of how these water-based beverage patterns differ in overall diet quality and how this understanding may help consumers achieve healthier dietary patterns. Additionally, we are not aware of any studies that have examined aspects of diet quality among consumers of the broader ZCUB category including unsweetened water, coffee, and tea.

The purpose of this study is to quantify measures of select nutrient intakes and diet quality among consumers of following two categories of water-based low- or no-calorie beverage options compared to consumers of SSB: (1) ZCUB including unsweetened water, coffee and tea, and (2) LNCSB. These comparisons, made across three life stages (i.e., children, adults, and older adults) and using data representative of the US population, will provide information on alternative water-based beverage options to SSB and may play a role in helping Americans meet dietary guidance recommendations.

## 2. Materials and Methods

### 2.1. Study Population Sample

This cross-sectional study was conducted with data collected in the combined 2009–2010, 2011–2012, 2013–2014, and 2015–2016 NHANES [13] and the dietary recall component known as What We Eat in America (WWEIA 2009–2016). The NHANES are designed to provide nationally representative nutrition and health data and prevalence estimates for nutrition and health status measures in the US [14]. Approval for the NHANES data collection was provided by the National Center for Health Statistics (NCHS) Research Ethics Review Board. 

In the WWEIA component of NHANES, detailed information on all foods and beverages consumed by participants in the previous 24-hour time period (midnight to midnight) was collected via two dietary recalls. The first recall was conducted in-person and the second dietary recall was administered by telephone three to ten days after the first dietary interview, but not on the same day of the week as the first interview. Trained interviewers conducted both recalls and data were collected using the US Department of Agriculture’s (USDA’s) Automated Multiple-Pass Method (AMPM), a 5-step method with multiple memory cues to facilitate more complete collection of all foods and beverages consumed [15]. The sample for this analysis was limited to non-breastfeeding females and males ages 2 years and older who provided a reliable dietary recall meeting the minimum criteria as determined by NCHS on Day 1 of the data collection (*n* = 32,959). The NHANES participants were categorized into three life stage groups based on their age: children (2–18 years), adults (19–64 years), and older adults (65+ years).

### 2.2. Classification of Water-Based Beverages

All beverages as reported consumed in WWEIA (2009–2016) were reviewed to identify water-based beverages of interest for the analyses, namely ZCUB, LNCSB, and SSB. In the NHANES dietary recall, participants report each food and beverage consumed, including consumption of more than one food or beverage items in a specific combination such as a sandwich, salad or beverage. The WWEIA data files capture each specific combination reported with an identifying variable for the combination type (e.g., a sandwich, salad or beverage) and a combination number to distinguish unique combinations reported throughout the day. In the current analyses, all beverage combinations as well as beverages not in a combination were reviewed to capture the identity of beverages as actually consumed. Any beverage or beverage combination containing a component other than a LNCSB, unsweetened water (which would include “carbonated or still” and either “flavored or unflavored” unsweetened waters), unsweetened coffee, unsweetened tea, SSB, or a sweetener (either caloric or a LNCS) was not considered to be a beverage of interest, and therefore, was not captured in the ZCUB, LNCSB, or SSB category. For example, brewed coffee with sugar and cream was not classified as an SSB due to the presence of cream. Lemon or lime juice was reported in some beverage combinations; this addition was assumed to provide flavor and was not a reason to exclude that beverage when classifying beverage combinations as noted above.

Based on this detailed review, the USDA food hierarchy, WWEIA food categories for beverages, food descriptions for each code, and nutrient concentration data (energy, total sugars, added sugars, and carbohydrates), all water-based beverages and beverage combinations of interest were identified and classified as a ZCUB, LNCSB, or SSB. All other beverages and beverage combinations were considered to be “other beverages”. A description of beverages captured in the ZCUB, LNCSB, and SSB categories is presented in Table 1.

### 2.3. Classification of Water-Based Beverage Consumers

Two independent analyses were conducted separately for (1) ZCUB and (2) LNCSB categories. Within each analysis, NHANES participants were assigned to four mutually exclusive beverage consumer groups, depending on whether they consumed SSB or the alternative water-based beverage options (i.e., ZCUB or LNCSB) on Day 1 of the data collection, irrespective of what other beverages they may have consumed. A summary of the water-based beverage consumer groups within each analysis is presented in Table 2. 

### 2.4. Nutrient Intakes

Intakes of added sugar, total sugar, and carbohydrates from the total diet were based on dietary intake data reported for Day 1 of the survey [13]. Since added sugar was not reported in the total nutrient intake datafiles, total daily intakes of added sugars (in grams) were derived from Day 1 dietary intakes and the Food Patterns Equivalents Database (FPED) [16] developed by the USDA that translates each food into food components assuming 4.2 grams added sugar per teaspoon.

### 2.5. Diet Quality

The most recent HEI-2015, designed to measure conformance with the 2015–2020 DGA, was selected as the basis for assessing diet quality [17]. Two methods were used to estimate the HEI-2015 component and overall scores. The first method, the population ratio method [18], was applied to estimate the mean HEI-2015 scores for the total study population by beverage consumer group for comparisons across survey cycles, and for comparisons of beverage consumer groups by life stage. The second method, the simple HEI scoring algorithm [18], was used to estimate the total and component HEI-2015 scores at the individual NHANES participant level for comparing scores across beverage consumer groups adjusting for the effect of study participant characteristics and survey cycle.

### 2.6. Characteristics of Study Population

Study participant characteristics selected a priori for inclusion in the analysis included: total energy intake, age, sex, race/ethnicity (“Mexican American”, “other Hispanic”, “white, non-Hispanic”, “black, non-Hispanic”, “other”), household reference person’s education level (high school graduate and below, some college and above), household poverty to income ratio (<130%, 130–300%, >300%), whether on a special diet (yes, no), and body mass index (BMI) category (underweight, normal weight, overweight, obese).

### 2.7. Statistical Analyses

The ZCUB (Analysis #1) and LNCSB (Analysis #2) analyses were conducted independently as two separate assessments using the same methods. Characteristics of the NHANES participants in each beverage consumer group were summarized and compared using Pearson Chi-square test for the categorical variables and the Wald test, with Bonferroni adjusted *p*-values for multiple comparisons, for the continuous variables. Linear regression methods were used to compare nutrient intakes across survey cycles for participants in each of the four beverage consumer groups. Bivariate analyses to compare participants in the four beverage consumer groups with respect to mean nutrient intakes and overall HEI-2015 score (derived using the population ratio method) were conducted by life stage (2–18 years, 19–64 years, and 65+ years). Pair-wise comparisons between each beverage consumer group (Table 2) versus “SSB only” consumers were conducted using the Wald test with Bonferroni adjusted *p*-values. Multivariate linear regression models were used to compare mean nutrient intakes and overall HEI-2015 scores (derived using the simple HEI scoring algorithm) among the beverage consumer groups with “SSB only” consumers as the reference group, adjusting for study participant characteristics, energy intake (nutrients only), and survey cycle. Generalized linear models (GLMs) [19] with a gamma distribution and log-link function were used to compare the mean HEI-2015 component scores among the beverage consumer groups with “SSB only” consumers as the reference group adjusting for study participant characteristics and survey cycle. These models do not require that the dependent variable be normally distributed and have been used to analyze the component HEI scores with other parameters [8,20]. 

All statistical analyses except the derivation of the HEI-2015 scores, were performed using STATA (version 12.1, 2014, StataCorp LP, College Station, TX, USA). The HEI scores were derived using SAS macros developed by the National Cancer Institute (NCI) [18]. All analyses used the statistical weights provided in NHANES to account for oversampling, survey non-response, and post-stratification and estimates of the standard errors (SE) and confidence intervals were design adjusted. Statistical significance was set at *p* < 0.05. 

## 3. Results

### 3.1. Analysis #1: ZCUB

#### 3.1.1. Characteristics of Study Population

All study population characteristics were significantly different among the ZCUB consumer groups (*p <* 0.0001). The “ZCUB only” consumers were more likely to be older, female, and white as well as from a household with higher income and more educated when compared to “SSB only” consumers among the US population (2+ years) (Table 3). Mean total energy intake was 383 kcals/day and 367 kcals/day lower among “ZCUB only” and “Neither” consumers, respectively, compared to “SSB only” consumers.

#### 3.1.2. Nutrient Intakes

Statistically significant trends were detected among “Both” and “ZCUB only” consumers (Figure 1a–c). Among “Both” consumers, mean total sugar intake decreased from 138 g/day (SE = 2.0 g/day) to 128 g/day (SE = 3.1 g/day) (*p*-trend < 0.0010), mean added sugar intake decreased from 100 g/day (SE = 1.8 g/day) to 92 g/day (SE = 2.5 g/day) (*p*-trend = 0.0050), and mean total carbohydrate intake decreased from 285 g/day (SE = 3.3 g/day) to 273 g/day (SE = 3.6 g/day) (*p*-trend = 0.0020) from 2009–2010 to 2015–2016, respectively. Among “ZCUB only” consumers, mean total sugar intake decreased from 88 g/day (SE = 1.2 g/day) to 79 g/day (SE = 1.3 g/day) (*p*-trend < 0.0001), mean added sugar intake decreased from 41 g/day (SE = 1.0 g/day) to 38 g/day (SE = 0.9 g/day) (*p*-trend < 0.0040), and mean total carbohydrate intake decreased from 225 g/day (SE = 2.6 g/day) to 210 g/day (SE = 2.6 g/day) (*p*-trend < 0.0001) from 2009–2010 to 2015–2016, respectively. “SSB only” and “Neither” consumers also showed an overall decline in mean total sugar, added sugar, and total carbohydrate intakes from 2009–2010 to 2015–2016 but trends were not statistically significant, most likely due to low statistical power.

Mean sugar (total and added) and carbohydrate intakes were statistically significantly lower among “ZCUB only” and “Neither” consumers across all life stages when compared pair-wise to “SSB only” consumers in unadjusted analyses (Table 4). The “ZCUB only” and “Neither” consumers on average consumed at least 50 grams less total sugar, 24 grams less added sugar, and 40 grams less total carbohydrates compared to “SSB only” consumers. The largest difference in mean added sugar intake was observed among adult “ZCUB only” when compared to “SSB only” with a difference of 107 g/day or approximately a 73% relative difference in intake. Similarly, compared to “SSB only” consumers, differences in mean total sugar and carbohydrate reductions were greatest among the “ZCUB only” consumers.

#### 3.1.3. Diet Quality

There were no statistically significant trends among “SSB”, “Both”, and “ZCUB only” consumers’ HEI-2015 score (Figure 1d). Diet quality was significantly higher, as measured by the overall HEI-2015 score, in “ZCUB only” and “Neither” consumers when compared to “SSB only” consumers (Table 4). These significantly higher scores were attenuated in adjusted analyses but remained statistically significant (Table 5). The highest overall diet quality score was observed among adults in the “ZCUB only” consumer group with an adjusted increase in the mean score of 9.2 points while older adults and children were 7.4 points and 7.2 points higher, respectively. Among the individual component scores, all life stages and “ZCUB only” or “Neither” consumers had at least double the added sugar score (in other words, less added sugar) when compared to “SSB only” consumers (Appendix A). The 9.2 point higher score observed among adult “ZCUB only” consumers in the adjusted analyses can be largely explained by the added sugar component. In analyses adjusted for participant characteristics, the relative difference in the added sugar component score of the HEI-2015 in adults 19–64 years was 2.6 (SE = 0.09), indicating that “ZCUB only” consumers’ added sugar score was 2.6 times higher than “SSB only” consumers’ score (Appendix A). Added sugar scores exceeded 80% (i.e., 8 out of 10 points) among all life stages in the “ZCUB only” consumer groups. “ZCUB only” or “Neither” consumers had lower component sodium and saturated fat scores (in other words, more sodium or saturated fat) compared to “SSB only” consumers, but these lower scores were offset in part by the higher added sugar scores resulting in an overall improvement in diet quality (Appendix A). 

In analyses adjusted for study population characteristics, survey cycles, and energy intake, the statistically significant lower mean total intakes of sugar, added sugar, and carbohydrates and higher mean HEI-2015 scores remained, though differences were slightly attenuated, when comparing “ZCUB only” and “Neither” consumer groups to “SSB only” consumers (Table 5). 

### 3.2. Analysis #2: LNCSB

#### 3.2.1. Characteristics of Study Population

All study population characteristics were significantly different among the LNCSB consumer groups (*p* < 0.0001). The “LNCSB only” consumers were more likely to be older, female, and white as well as from households with a higher income and more educated when compared to “SSB only” consumers among the US population (2+ years) and were more likely to be overweight or obese and on a special diet (Table 6). The “LNCSB only” and “Neither” consumers also had a lower total energy intake compared to “SSB only” consumers; “LNCSB only” consumers had 262 kcal/day fewer while the “Neither” consumers had 333 kcal/day fewer. 

#### 3.2.2. Nutrient Intakes

Statistically significant declining trends in the intakes of total and added sugars and carbohydrates were detected among “SSB only” and “Neither” consumers in the total study population (2+ years), while non-significant declining trends were observed among “LNCSB only” consumers and fluctuations in intakes were observed within the “Both” consumer group (Figure 2a–c). Among “SSB only” consumers, mean total sugar intake decreased from 148 g/day (SE = 1.9 g/day) to 134 g/day (SE = 2.5 g/day) (*p*-trend < 0.0001), mean added sugar intake decreased from 109 g/day (SE = 1.8 g/day) to 99 g/day (SE = 2.1 g/day) (*p*-trend = 0.0010), and mean total carbohydrate intake decreased from 291 g/day (SE = 3.1 g/day) to 275 g/day (SE = 3.1 g/day) (*p*-trend < 0.0001) from 2009–2010 to 2015–2016, respectively. Among “Neither” consumers, mean total sugar intake decreased from 94 g/day (SE = 1.3 g/day) to 80 g/day (SE = 1.3 g/day) (*p*-trend < 0.0001), mean added sugar intake decreased from 42 g/day (SE = 1.3 g/day) to 37 g/day (SE = 0.9 g/day) (*p*-trend < 0.0001), and mean total carbohydrate intake decreased from 229 g/day (SE = 2.6 g/day) to 211 g/day (SE = 3.0 g/day) (*p*-trend < 0.0001) from 2009–2010 to 2015–2016, respectively. “LNCSB only” consumers had the lowest mean total sugar intake in 2009–2010 at 80 g/day (SE = 2.6 g/day) which declined to 75 g/day (SE = 3.2 g/day) in 2015–2016 (*p*-trend = 0.2450).

Across all life stage groups, “LNCSB only” and “Neither” consumers had statistically significant lower mean intakes of added sugar, total sugar, and carbohydrates when compared to “SSB only” consumers in unadjusted analyses (Table 7). Among children and older adults, mean added sugar intake among these two consumer groups were generally half that of the “SSB only” consumers while mean total sugar intake were about one-third the mean intake of “SSB only” consumers. The largest absolute reduction in total sugar intake relative to “SSB only” consumers was observed among the “LNCSB only” consumers. There were no significant differences between the “Both” and “SSB only” consumer groups among children and older adults with the exception of a reduction in mean added sugar intake among the older adults. Among adults, all beverage consumer groups had statistically significant lower mean added sugar, total sugar, and carbohydrate intakes when compared to “SSB only” consumers and again, the largest difference in total sugar intake was among the “LNCSB only” consumers with a mean intake of 79 g/day (SE = 1.9 g/day) relative to 149 g/day (SE = 1.6 g/day) for “SSB only” consumers.

#### 3.2.3. Diet Quality

There were no statistically significant trends among “SSB”, “Both”, and “LNCSB only” consumers’ HEI-2015 score (Figure 2d). Diet quality as measured by the HEI-2015 was significantly higher in the “LNCSB only” and “Neither” consumers when compared pair-wise to “SSB only” among adults and older adults, and among children in the “Neither” group (Table 7). After adjustment for study population characteristics and survey cycle, adult “LNCSB only” consumers were on average 3.9-points higher than “SSB only” consumers (Table 8). The highest overall HEI-2015 scores among all life stages were observed among “Neither” consumers with scores 6.1 to 7.6 points higher than “SSB only” consumers in adjusted analyses (Table 8). In all life stage groups the higher overall HEI-score was largely driven by the added sugar component based on analysis of the individual HEI-2015 components (Appendix A). Added sugar scores were 80% (i.e., 8 out of 10 points) among children (Appendix A) while adults were above 90% in unadjusted analyses (Appendix A). Therefore, without SSB consumption, added sugar scores were close to two times higher (i.e., lower added sugar) among children and more than doubled among adults in the “LNCSB only” consumer group; the largest relative and absolute higher component score observed among the beverage consumer groups. In analyses adjusted for participant characteristics, the relative difference in the added sugar scores was 1.7 (SE = 0.05), 1.9 (SE = 0.04), and 1.7 (SE = 0.05) among children, adults, and older adults, respectively, indicating that “LNCSB only” consumers’ added sugar score remained close to two times higher than “SSB only” consumers (Appendix A). The “LNCSB only” or “Neither” consumers had slightly lower sodium and saturated fat scores compared to “SSB only” consumers; however, both the absolute and relative reduction in score was smaller compared to the higher added sugar components as well as the other dietary components including fruits and dairy (Appendix A). 

In adjusted analyses, the statistically significant lower mean added sugar, total sugar, and carbohydrate intakes and higher HEI-2015 scores remained but were slightly attenuated when comparing “LNCSB only” and “Neither” consumers to “SSB only” consumers (Table 8). 

## 4. Discussion

Dietary guidance in the US encourages consumers to reduce consumption of SSB as a strategy to lower added sugar intakes and in turn improve overall diet quality. Using recent and nationally representative data for individuals 2 years of age and older, this study quantifies differences in mean sugar and carbohydrate intakes and diet quality comparing consumers of alternative water-based beverage options including ZCUB and separately LNCSB to SSB consumers, with adjustments for participant characteristics. Findings from these two independent analyses provide quantitative data to allow for a better understanding of the value ZCUB or LNCSB bring in helping Americans across life stages meet dietary guidance recommendations.

Results from this study show that compared to SSB consumers, ZCUB consumers have significantly lower mean dietary intakes of total sugars, added sugars, and carbohydrates, and a significantly higher diet quality as measured by the HEI-2015. In addition, consumers who reported both SSB and ZCUB consumption had significantly lower mean intakes of added sugars and higher diet quality. Similar results were observed in the LNCSB analyses where comparisons of adult LNCSB consumers to SSB consumers showed significantly lower mean dietary intakes of total sugars, added sugars and carbohydrates and higher diet quality among LNCSB consumers. Compared to SSB consumers, adult and older adult consumers of both SSB and LNCSB had lower intakes of added sugars, but only older adult consumers of both beverage categories had higher diet quality. The analyses suggest that replacement of some SSB with either ZCUB or LNCSB may also help improve select dietary measures.

Findings in both the ZCUB and LNCSB analyses clearly show that dietary patterns with alternative water-based beverage options are higher quality and contain less sugar and carbohydrate than patterns including SSB, thus showing that on the day of reported consumption, participants that did not consume a SSB had lower mean total and added sugar intakes from all dietary sources. Differences in mean added sugar intake between “ZCUB only” consumers compared to “SSB only” consumers were correlated with the lack of SSB consumption. For example, added sugar intake among children “ZCUB only” consumers was 59 g/day lower compared to “SSB only” consumers, corresponding to 683 mL/day lower SSB consumption (i.e., less than 24 fluid ounces) (see Table 4). Further, the HEI component analysis supports this observation given “SSB only” consumers had a mean added sugar component score of 3.3 while “ZCUB only” consumers were at 8.5 (see Appendix A). Comparable observations can be made among adults and older adults. Similarly, the mean added sugar intake among “LNCSB only” adult consumers with no SSB consumption was 72 g/day lower compared to “SSB only” consumers with reported mean SSB consumption of 865 mL/day (Table 7). 

Findings from the current study corroborate those from other published studies, showing a higher diet quality or lower intakes of calories, sugar, and/or carbohydrates when comparing consumers of either unsweetened beverages including water [8,10,21] or low and no-calorie sweetened beverages [1,11,22] to SSB consumers. However, definitions of beverages and beverage consumer groups are not fully consistent across the literature, which makes direct comparison difficult. In this study, both the sweetened and unsweetened beverages considered in the primary categorizations are water-based beverages with few nutrients other than possibly energy. Classification of beverage consumers also accounted for the addition of either caloric or non-caloric sweetening components (e.g., sugar added to tea or low- and no-calorie sweeteners added to coffee, respectively) to beverages prior to consumption, and is a classification not routinely considered in analyses of beverage patterns [23]. Nevertheless, the addition of sugar and/or a dairy cream to coffee and tea prior to consumption is a common practice in the US [24]. Other combinations also may have modified the overall beverage profile; therefore, it was necessary to consider combinations to accurately distinguish among sweetened, unsweetened, and other beverages as consumed by the US population.

In the case of SSB, the water-based beverages selected for inclusion typically provide only energy in the form of added sugars. The alternative beverages selected for the analysis provide few nutrients (including energy) as they are inherently calorie-free beverages (e.g., water, coffee or tea), or contain few nutrients due to the use of a high-intensity sweetener. Other definitions of sweetened beverages have included flavored milks, flavored grain-based beverages, or combinations of coffee with a milk/cream and coffee; these beverages are sweetened and a source of sugars in the diet, though they are also a potential source of other nutrients. In the current analyses, these beverage and beverage combinations were included in the “Other” beverage category. 

In addition to varying definitions of beverages across the literature, it is also important to consider beverage consumer classification methods in the current study in the interpretation of results. Given the objectives of this study and the methods used to define beverage consumers within the ZCUB and LNCSB analyses, the comparator group of “SSB only” as well as the “Neither” consumer group is comprised of different individuals within the two analyses. Specifically, the “Neither” group in the LNCSB analyses includes unsweetened water and coffee/tea consumers who did not consume any SSB; while in the ZCUB analyses, the “Neither” group includes LNCSB consumers who did not consume any SSB. Similarly, the “SSB only” group in the LNCSB analyses includes SSB consumers without excluding consumers that consume both SSB and unsweetened water and coffee/tea while in the ZCUB analyses, the “SSB only” group includes SSB consumers without excluding consumers who consume both SSB and LNCSB. Therefore, the results of the ZCUB analyses and the LNCSB analyses are not directly comparable and should be viewed as two independent analyses. 

Classification of participants into beverage consumer groups was based on self-report of any amount of a beverage identified as either a ZCUB, LNCSB, or SSB; individuals reporting no intake of either of the two relevant beverage categories in an analysis were classified as “Neither”. There was no threshold consumption amount required to place a participant in each group. Therefore, an individual who reported 10 mL/day or 1000 mL/day of an SSB would be defined as an SSB consumer. Inclusion of individuals with small or incidental intakes of ZCUB, LNCSB, or SSB could result in the attenuation of the associations observed. However, the differences in intake and diet quality among the beverage consumer groups were relatively large and highly statistically significant, reducing the potential for a missed association due to misclassification of beverage consumers. 

The analyses presented in this study were based on cross-sectional data and therefore, conclusions are limited to quantified associations without the ability to assess temporality or causality. Further, a single 24-hour dietary recall record was used to both classify beverage consumers as well as estimate nutrient intakes and diet quality. While this single day of recall may create misclassification bias by failing to identify all typical consumers of these products and not just those on the day of data collection, 24-hour recall is a dietary assessment method known to provide valid estimates of mean population intakes, upon which our conclusions are based [25]. The WWEIA dietary recall data were collected via self-report methods subject to (under-)reporting bias resulting in measurement error. Classification of participants into beverage consumer groups was based on self-report of consumption of a beverages and categorization of the beverages as consumed. The multi-pass method used as part of the in-person interview attempts to reduce this error and has been shown to reduce bias in estimation of total energy intakes [15]. In the case of SSB, the social desirability of these products has declined in recent years, and therefore, participants may be inclined to under-report use, which would attenuate the relationships we observed in the current study. Further, while dietary data for children <12 years were collected with the assistance of an adult proxy, research has shown that the 24-hour dietary recall with multi-pass is valid for children and proxies [26]. Although participant characteristics including demographic, socioeconomic, and anthropometric measures that may affect the associations observed in the current study were included in the adjusted analyses, residual confounding may remain. Strengths of these analyses include the use of four cycles of NHANES dietary data representative of the US population, a large study population, adjustment for energy intake and potential confounding factors present among the participants, and the use of statistical tests adjusted for multiple comparisons. To our knowledge, this is one of the first analyses that includes all three life stages from children to older adults and compares nutrient intakes and diet quality from consumption of ZCUB and LNCSB to SSB in a population representative of the US. 

## 5. Conclusions

Our findings show that consumers of zero-calorie unsweetened or low- and no-calorie sweetened water-based beverages have lower mean intakes of total sugar, added sugar, and carbohydrates and higher mean diet quality when compared to SSB consumers. Higher diet quality was driven in large part by higher added sugar (i.e., lower added sugar intakes) component HEI scores. These significant findings remained in analyses adjusted for demographic, socioeconomic, and anthropometric factors among study participants and total energy intake. Further, while the absolute and relative magnitude of the differences varied, these findings were consistent among all life stages. The current cross-sectional analyses present significant associations between ZCUB or LNCSB consumption and reduced sugar and carbohydrate intakes and higher diet quality when compared to SSB consumption. Findings from this study suggest that replacement of SSB with ZCUB or LNCSB could help Americans achieve dietary recommendations on the added sugars component of the HEI and improve overall diet quality.

## Figures and Tables

**Figure 1 nutrients-11-00314-f001:**
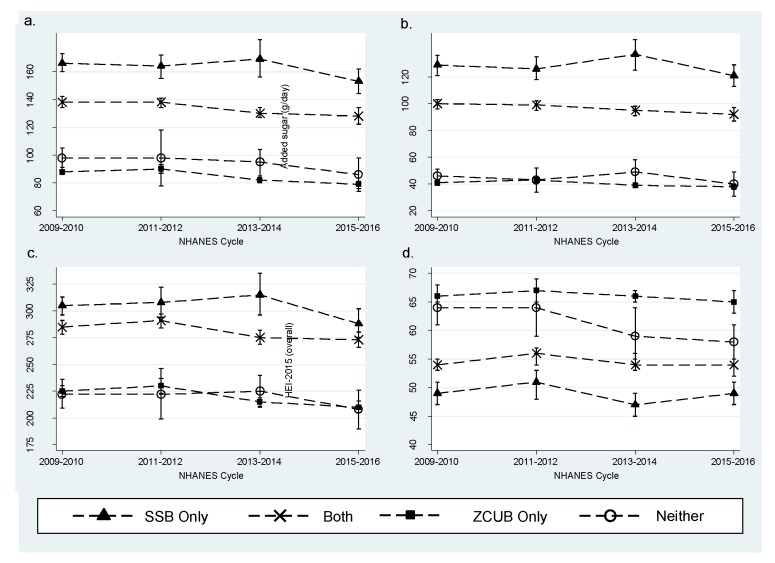
Nutrient intakes and diet quality of the US population (2+ years) by ZCUB consumer group, NHANES 2009–2016. (**a**) Mean Day 1 intake of total sugars (g/day) from the total diet; (**b**) mean Day 1 intake of added sugars (g/day) from the total diet; (**c**) mean Day 1 intake of total carbohydrates (g/day) from the total diet; (**d**) HEI-2015 overall score calculated using the population ratio method and Day 1 intakes.

**Figure 2 nutrients-11-00314-f002:**
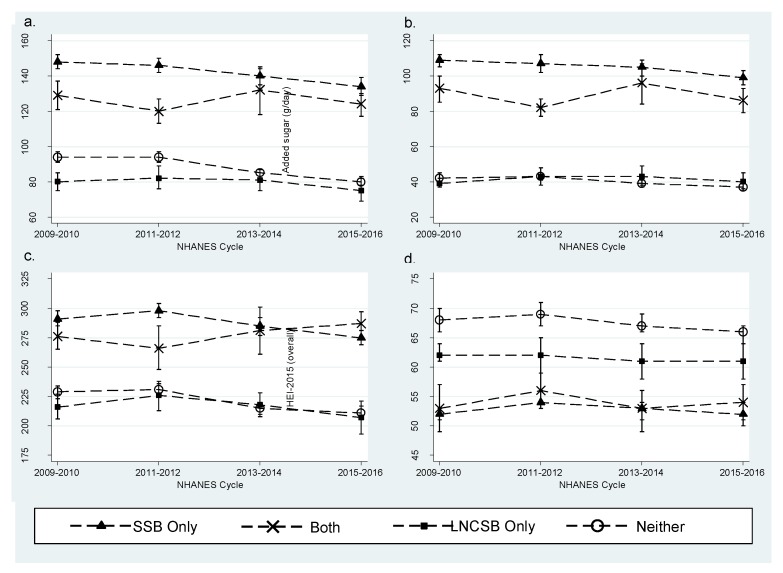
Nutrient intakes and diet quality of the US population (2+ years) by LNCSB consumer group, NHANES 2009–2016. (**a**) Mean Day 1 intake of total sugars (g/day) from the total diet; (**b**) mean Day 1 intake of added sugars (g/day) from the total diet; (**c**) mean Day 1 intake of total carbohydrates (g/day) from the total diet; (**d**) HEI-2015 overall score calculated using the population ratio method and Day 1 intakes.

**Table 1 nutrients-11-00314-t001:** Classification of water-based beverages in analyses.

Water-Based Beverage Category	Description of Water-Based Beverages
Zero calorie unsweetened beverages (ZCUB)	Tap water, unsweetened bottled water (including unsweetened flavored, carbonated, enhanced, or fortified), unsweetened coffee (includes brewed, instant, espresso), unsweetened tea (includes leaf, instant, brewed, hot, iced, bottled)
Low- or no-calorie sweetened beverages (LNCSB)	Soft drinks, water (including sweetened flavored, carbonated, enhanced, or fortified), and fruit, sport, and energy drinks identified as diet/low calorie/no sugar/sugar free/zero and containing no added sugars; coffee and tea presweetened with a low or no calorie sweetener (LNCS) or sweetened with an LNCS prior to consumption and containing no other components
Sugar-sweetened beverages (SSB)	Soft drinks; calorically sweetened water (including sweetened flavored, carbonated, enhanced, or fortified); fruit, sport and energy drinks (including light, low calorie, reduced sugar, and lo carb drinks containing added sugars); calorically sweetened fruit juice; calorically sweetened coconut water; nectar; coffee or tea presweetened with sugar or sweetened with sugar prior to consumption and containing no other components

**Table 2 nutrients-11-00314-t002:** Definition of water-based beverage consumer groups.

Classification for Analysis	Water-Based Beverage Consumer Group (*N* = Number of Consumers)	Description	
Analysis #1: ZCUB		SSB	ZCUB
	“SSB only” (*N* = 4637)	Yes	No
	“Both” (*N* = 14,414)	Yes	Yes
	“ZCUB only” (*N* = 12,452)	No	Yes
	“Neither” (*N* = 1456)	No	No
Analysis #2: LNCSB		SSB	LNCSB
	“SSB only” (*N* = 17,554)	Yes	No
	“Both” (*N* = 1497)	Yes	Yes
	“LNCSB only” (*N* = 3215)	No	Yes
	“Neither” (*N* = 10,693)	No	No

**Table 3 nutrients-11-00314-t003:** Characteristics of study population by Zero-Calorie Unsweetened Beverage (ZCUB) and Sugar-Sweetened Beverage (SSB) consumer groups, NHANES 2009–2016.

	Total Study Population	Water-Based Beverage Consumer Group				*p*-Value ^a^
		SSB Only	Both	ZCUB Only	Neither	
Unweighted *N*	32,959	4637	14,414	12,452	1456	
Age (years) ^b^	38.4 (0.30)	31.8 (0.41)	35.4 (0.35)	42.9 (0.43)	40.5 (0.91)	<0.0001
Sex ^c^						
Male	49.0	56.4	51.4	44.4	52.2	<0.0001
Female	51.0	43.6	48.6	55.6	47.9	
Race/ethnicity ^c^						
Mexican American	10.3	12.4	13.2	7.1	8.7	<0.0001
Other Hispanic	6.3	7.9	7.4	4.8	5.8	
White, non-Hispanic	63.1	54.9	57.0	70.7	68.7	
Black, non-Hispanic	12.1	19.4	14.6	7.8	10.9	
Other race	8.2	5.4	7.8	9.6	5.9	
Total energy intake (kcal/day) ^b^	2086 (7.5)	2303 (27.5)	2215 (12.4)	1920 (11.6)	1936 (42.7)	<0.0001
Household poverty income ratio ^c^						
<130%	26.3	41.0	29.3	19.2	28.2	<0.0001
130–300%	28.9	31.3	30.8	26.4	29.4	
>300%	44.9	27.7	39.9	54.5	42.5	
Household reference education ^c^						
≤ high school	37.9	54.1	41.2	29.8	44.5	<0.0001
> high school	62.1	45.9	58.8	70.2	55.5	
BMI category ^c^						
Underweight	12.2	16.0	12.4	10.3	18.7	<0.0001
Normal	30.4	32.6	31.1	29.8	23.6	
Overweight	27.5	24.3	26.3	29.7	25.1	
Obese	29.9	27.2	30.2	30.2	32.5	
On a special diet? ^c^						
Yes	13.0	4.9	9.6	18.1	17.3	<0.0001
No	87.0	95.1	90.4	81.9	82.7	

^a^ Wald test with Bonferroni adjusted *p*-values for multiple comparisons (continuous variables) or Pearson Chi-square test (categorical variables); ^b^ Means (SE); ^c^ Percent (%) within beverage group.

**Table 4 nutrients-11-00314-t004:** Daily beverage consumption (mL/day), nutrient intakes (g/day) (from total diet), and diet quality across Zero-Calorie Unsweetened Beverage (ZCUB) and Sugar-Sweetened Beverage (SSB) consumer groups and life stages, NHANES 2009–2016.

	SSB Only	Both	ZCUB Only	Neither
	Mean (SE)			
**2–18 years**				
Unweighted *N*	2105	5526	3564	583
SSB ^a^ (mL/day)	683 (28.2)	516 (11.1)	0	0
ZCUB ^b^ (mL/day)	0	747 (18.1)	808 (25.5)	0
Other beverage ^c^ (mL/day)	360 (12.9)	297 (9.4)	458 (11.3)	897 (121.3)
Added sugar (g/day)	101 (2.5)	89 (1.3) *	42 (0.8) *	48 (3.6) *
Total sugar (g/day)	141 (2.9)	129 (1.4) *	94 (1.2) *	117 (7.4) *
Carbohydrates (g/day)	270 (4.7)	270 (2.1)	220 (2.1) *	224 (9.0) *
HEI-2015	48.8 (0.65)	51.0 (0.51) *	60.8 (0.65) *	59.7 (0.95) *
**19–64 years**				
Unweighted *N*	2176	7340	6343	595
SSB ^a^ (mL/day)	1196 (31.7)	742 (11.3)	0	0
ZCUB ^b^ (mL/day)	0	1442 (25.0)	1834 (26.3)	0
Other beverage ^c^ (mL/day)	792 (50.8)	604 (17.5)	932 (19.40	1877 (61.3)
Added sugar (g/day)	147 (3.3)	102 (1.2) *	40 (0.7) *	47 (3.0) *
Total sugar (g/day)	179 (3.6)	138 (1.4) *	83 (0.9) *	89 (3.9) *
Carbohydrates (g/day)	328 (5.2)	292 (2.2) *	224 (1.8) *	226 (6.2) *
HEI-2015	48.0 (0.75)	54.3 (0.46) *	66.4 (0.48) *	59.1 (1.38) *
**65+ years**				
Unweighted *N*	356	1548	2545	278
SSB ^a^ (mL/day)	739 (51.5)	511 (19.4)	0	0
ZCUB ^b^ (mL/day)	0	1089 (45.0)	1368 (38.3)	0
Other beverage ^c^ (mL/day)	636 (41.6)	476 (25.7)	672 (15.5)	1368 (59.4)
Added sugar (g/day)	90 (4.6)	80 (1.9)	39 (0.9) *	37 (1.9) *
Total sugar (g/day)	127 (5.4)	119 (2.4)	82 (1.2) *	81 (3.6) *
Carbohydrates (g/day)	247 (8.3)	245 (4.0)	203 (2.2) *	200 (6.1) *
HEI-2015	56.6 (1.44)	61.5 (0.69) *	68.1 (0.71) *	67.2 (2.12) *

Note: beverage consumption reported in milliliters/day (mL/day) assuming density of water (i.e., 1 g/mL). ^a^ SSB include: soft drinks; calorically sweetened water (including sweetened flavored, carbonated, enhanced, or fortified); fruit, sport, and energy drinks (including light, low calorie, reduced sugar, and low carb drinks containing added sugars); calorically sweetened fruit juice; calorically sweetened coconut water; nectar; coffee or tea presweetened with sugar or sweetened with sugar prior to consumption and containing no other components; ^b^ ZCUB include: Tap water, unsweetened bottled water (including unsweetened flavored, carbonated, enhanced, fortified), unsweetened coffee (includes brewed, instant, espresso), unsweetened tea (includes leaf, instant, brewed, hot, iced, bottled); ^c^ Other includes: all other beverages and beverage combinations not included in the SSB or ZCUB categories (e.g., milk beverages, 100% fruit juices, coffee/tea with creamer); * Indicates statistically significant difference when compared to “SSB only”; adjusted Wald test with Bonferroni adjusted *p*-values for multiple comparisons.

**Table 5 nutrients-11-00314-t005:** Associations between nutrient intakes (from total diet) and diet quality across Zero-Calorie Unsweetened Beverage (ZCUB) and Sugar-Sweetened Beverage (SSB) consumer groups and life stages, NHANES 2009–2016.

	SSB Only	Both	ZCUB Only	Neither
	Coefficient β (SE)			
**2–18 years**				
Added sugar (g/day) ^a^	Ref	−13.6 (1.97) *	−48.5 (1.82) *	−41.6 (2.62) *
Total sugar (g/day) ^a^	Ref	−14.0 (2.02) *	−34.5 (2.19) *	−12.7 (4.34) *
Carbohydrates (g/day) ^a^	Ref	−5.5 (2.12) *	−20.4 (2.33) *	−16.6 (2.30) *
HEI-2015 ^b^	Ref	1.5 (0.45) *	7.2 (0.54) *	6.3 (0.80) *
**19–64 years**				
Added sugar (g/day) ^a^	Ref	−36.7 (2.53) *	−87.3 (2.60) *	−85.5 (3.21) *
Total sugar (g/day) ^a^	Ref	−31.5 (2.71) *	−72.4 (2.75) *	−72.3 (4.06) *
Carbohydrates (g/day) ^a^	Ref	−17.3 (2.64) *	−48.9 (2.63) *	−59.3 (4.14) *
HEI-2015 ^b^	Ref	2.9 (0.45) *	9.2 (0.55) *	5.0 (0.87) *
**65+ years**				
Added sugar (g/day) ^a^	Ref	−9.2 (3.97) *	−46.0 (3.87) *	−47.7 (4.03) *
Total sugar (g/day) ^a^	Ref	−7.3 (4.00)	−38.6 (4.02) *	−37.7 (5.00) *
Carbohydrates (g/day) ^a^	Ref	−2.6 (4.27)	−27.8 (4.25) *	−28.1 (5.58) *
HEI-2015 ^b^	Ref	3.0 (1.17) *	7.4 (1.08) *	5.4 (1.61) *

^a^ Linear regression models adjusted for energy intake, survey cycle, age, sex, race/ethnicity, BMI, household reference education, household poverty income ratio, and dieting status; ^b^ Linear regression models adjusted for survey cycle, age, sex, race/ethnicity, BMI, household reference education, household poverty income ratio, and dieting status; * Significant at the *p* < 0.05 level.

**Table 6 nutrients-11-00314-t006:** Characteristics of study population by Low/No-Calorie Sweetened Beverage (LNCSB) and Sugar-Sweetened Beverage (SSB) consumer groups, NHANES 2009–2016.

	Total Study Population	Water-Based Beverage Consumer Group				*p*-Value ^a^
		SSB Only	Both	LNCSB Only	Neither	
Unweighted *N*	32,959	17,554	1497	3215	10,693	
Age (years) ^b^	38.4 (0.30)	33.9 (0.30)	40.7 (0.88)	48.7 (0.50)	40.3 (0.46)	<0.0001
Sex ^c^						
Male	49.0	52.7	50.9	43.5	45.7	<0.0001
Female	51.0	47.3	49.1	56.5	54.3	
Race/ethnicity ^c^						
Mexican American	10.3	13.6	7.6	5.2	8.1	<0.0001
Other Hispanic	6.3	7.7	5.5	3.5	5.5	
White, non-Hispanic	63.1	54.8	71.3	81.7	65.9	
Black, non-Hispanic	12.1	16.4	9.1	5.2	9.3	
Other race	8.2	7.4	6.5	4.4	11.3	
Total energy intake (kcal/day) ^b^	2086 (7.5)	2234 (11.6)	2242 (28.1)	1972 (23.9)	1901 (11.9)	<0.0001
Household poverty income ratio ^c^						
<130%	26.3	33.3	20.2	13.7	22.6	<0.0001
130–300%	28.9	31.2	28.5	23.3	28.0	
>300%	44.9	35.5	51.4	63.0	49.4	
Household reference education ^c^						
<=high school	37.9	44.8	37.6	29.8	31.6	<0.0001
>high school	62.1	55.2	62.4	70.2	68.5	
BMI category ^c^						
Underweight	12.2	14.1	5.9	4.2	13.9	<0.0001
Normal	30.4	31.9	27.2	20.4	32.8	
Overweight	27.5	25.9	26.0	31.6	28.4	
Obese	29.9	28.2	40.9	43.9	24.9	
On a special diet? ^c^						
Yes	13.0	7.7	16.1	25.9	14.8	<0.0001
No	87.0	92.3	83.9	74.1	85.2	

^a^ Wald test with Bonferroni adjusted *p*-values for multiple comparisons (continuous variables) or Pearson Chi-square test (categorical variables); ^b^ Means (SE); ^c^ Percent (%) within beverage consumer group.

**Table 7 nutrients-11-00314-t007:** Daily beverage consumption (mL/day), nutrient intakes (g/day) (from total diet) and diet quality across Low/No-Calorie Sweetened Beverage (LNCSB) and Sugar-Sweetened Beverage (SSB) consumer groups and life stages, NHANES 2009–2016.

	SSB Only	Both	LNCSB Only	Neither
	Mean (SE)			
**2–18 years**				
Unweighted N	7203	428	445	3702
SSB ^a^ (mL/day)	558 (11.8)	586 (38.0)	0	0
LNCSB ^b^ (mL/day)	0	396 (29.6)	428 (23.4)	0
Other beverage ^c^ (mL/day)	836 (17.0)	812 (63.0)	986 (74.8)	1190 (28.0)
Added sugar (g/day)	92 (1.3)	94 (3.6)	49 (1.8) *	42 (0.9) *
Total sugar (g/day)	132 (1.4)	134 (4.6)	91 (2.8) *	97 (1.6) *
Carbohydrates (g/day)	269 (2.3)	279 (9.1)	223 (6.6) *	220 (2.4) *
HEI-2015	50.5 (0.44)	49.2 (1.66)	53.7 (1.87)	61.7 (0.56) *
**19–64 years**				
Unweighted N	8667	849	1922	5016
SSB ^a^ (mL/day)	865 (12.7)	633 (27.1)	0	0
LNCSB ^b^ (mL/day)	0	715 (30.1)	909 (23.1)	0
Other beverage ^c^ (mL/day)	1717 (33.30)	1545 (95.1)	1881 (37.8)	2654 (35.8)
Added sugar (g/day)	114 (1.5)	91 (2.3) *	42 (1.4) *	40 (0.8) *
Total sugar (g/day)	149 (1.6)	126 (2.8) *	79 (1.9) *	85 (1.0) *
Carbohydrates (g/day)	302 (2.2)	282 (6.1) *	223 (3.6) *	225 (1.9) *
HEI-2015	52.8 (0.47)	53.8 (1.08)	61.4 (0.86) *	67.9 (0.61) *
**65+ years**				
Unweighted N	1684	220	848	1975
SSB ^a^ (mL/day)	569 (20.3)	410 (28.9)	0	0
LNCSB ^b^ (mL/day)	0	589 (61.0)	646 (21.2)	0
Other beverage ^c^ (mL/day)	1361 (47.1)	1172 (101.9)	1375 (50.8)	1964 (44.3)
Added sugar (g/day)	83 (1.9)	71 (3.8) *	38 (1.6) *	39 (0.9) *
Total sugar (g/day)	122 (2.5)	111 (5.5)	76 (2.1) *	85 (1.3) *
Carbohydrates (g/day)	245 (4.0)	247 (8.2)	197 (3.8) *	206 (2.6) *
HEI-2015	60.7 (0.69)	61.4 (1.67)	64.5 (1.21) *	69.9 (0.88) *

Note: Beverage consumption reported in milliliters/day (mL/day) assuming density of water (i.e., 1 g/mL). ^a^ SSB include: soft drinks; calorically sweetened water (including sweetened flavored, carbonated, enhanced, or fortified); fruit, sport and energy drinks (including light, low calorie, reduced sugar, and low carb drinks containing added sugars); calorically sweetened fruit juice; calorically sweetened coconut water; nectar; coffee or tea presweetened with sugar or sweetened with sugar prior to consumption and containing no other components; ^b^ LNCSB include: Soft drinks, water (including non-calorically sweetened flavored, carbonated, enhanced, or fortified), and fruit, sport, and energy drinks identified as diet/low calorie/no sugar/sugar free/zero and containing no added sugars; coffee and tea presweetened with a LNCS or sweetened with a LNCS prior to consumption and containing no other components; ^c^ Other includes: all other beverages and beverage combinations not included in the SSB or LNCSB categories (e.g., milk beverages, 100% fruit juices, coffee/tea with creamer); * Indicates statistically significant difference when compared to “SSB only”; adjusted Wald test with Bonferroni adjusted *p*-values for multiple comparisons.

**Table 8 nutrients-11-00314-t008:** Associations between nutrient intakes (from total diet) and diet quality across Low/No-Calorie Sweetened Beverage (LNCSB) and Sugar-Sweetened Beverage (SSB) consumer groups and life stages, NHANES 2009–2016.

	SSB Only	Both	LNCSB Only	Neither
	Coefficient β (SE)			
**2–18 years**				
Added sugar (g/day) ^a^	Ref	−2.2 (3.30)	−37.6 (2.23) *	−37.6 (1.15) *
Total sugar (g/day) ^a^	Ref	−1.6 (4.15)	−32.1 (2.63) *	−19.6 (1.71) *
Carbohydrates (g/day) ^a^	Ref	2.8 (4.16)	−23.7 (3.10) *	−14.3 (1.66) *
HEI-2015 ^b^	Ref	−1.4 (1.06)	2.3 (0.94) *	6.5 (0.44) *
**19–64 years**				
Added sugar (g/day) ^a^	Ref	−21.0 (2.37) *	−61.6 (1.60) *	−59.6 (1.40) *
Total sugar (g/day) ^a^	Ref	−20.7 (2.64) *	−56.4 (1.49) *	−46.5 (1.66) *
Carbohydrates (g/day) ^a^	Ref	−13.6 (2.93) *	−41.6 (1.79) *	−35.7 (1.80) *
HEI-2015 ^b^	Ref	−0.2 (0.73)	3.9 (0.63) *	7.6 (0.41) *
**65+ years**				
Added sugar (g/day) ^a^	Ref	−15.1 (3.39) *	−40.0 (2.34) *	−40.6 (1.78) *
Total sugar (g/day) ^a^	Ref	−13.5 (5.00) *	−38.3 (2.68) *	−32.2 (2.09) *
Carbohydrates (g/day) ^a^	Ref	−6.2 (5.16)	−29.6 (3.39) *	−24.9 (2.51) *
HEI-2015 ^b^	Ref	2.5 (1.10) *	2.9 (0.81) *	6.1 (0.59) *

^a^ Linear regression models adjusted for energy intake, survey cycle, age, sex, race/ethnicity, BMI, household reference education, household poverty income ratio, and dieting status; ^b^ Linear regression models adjusted for survey cycle, age, sex, race/ethnicity, BMI, household reference education, household poverty income ratio, and dieting status; * Significant at the *p* < 0.05 level.

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
