# Peer review of "Comparisons of Nutrient Intakes and Diet Quality among Water-Based Beverage Consumers"

_nutrients, 2019, doi:10.3390/nu11020314_

Reviewer 1 Report

This study used NHANES 2009-2016 cross-sectional data to investigate the differences in intakes of added sugars, total sugars, carbohydrates, and diet quality among SSB, ZUCB, and LNCSB consumer groups in 3 life stages, and try to offer dietary recommendation. The follows are the review comments.

Introduction

Line 36: There might be a typo in the statement ‘coffee of tea with no added sweeteners’.

Lines 64-68: This statement seems to be long to be recognized.

Methods

Lines 125-126: It is not clear about the statement “the alternative water-based alternative beverage options (i.e., ZCUB or LNCSB)”.

Results

Lines 193-205: The results of Figure 1-d (HEI-2015 overall score) and Figure 2-d seem not to be reported.

Lines 218-220: The statement “mean total sugar and carbohydrate reductions” should be “the differences in mean total sugar and carbohydrate reductions”.

Lines 246-247: The statement “the relative difference in added sugar scores” is not clear. It could be revised as “ …the relative difference in HEI-2015 component scores for added sugar in adults aged 19-64 years”.

Line 275: The authors need to explain why the number for “SSB only” is 4,637 in Table 3 and 17,554 in Table 6. In Figure 2, statistically significant declining trends in the intakes of total and added sugars and carbohydrates were detected among “SSB only”, but in Figure 1, there were no such declining trends for “SSB only” consumers. In the discussion section (lines 420-429), the authors offered an explanation for this issue. Could the authors display the percentage distribution for the categories that are overlapped and not overlapped for the “SSB only” group in these two series of analyses.

Line 279: The subtitle should be “3.2.2. Nutrient Intakes”.

Line 346: There is an unbalanced quotes in this statement “LNCSB only consumers’ added”

Lines 362-363: “sugar-sweetened 362 beverages” can be abbreviated to be “SSB”.

Lines 374-376: This statement needs to cite references to denote “similar results”.

Lines 404-405: “the addition of sugar and/or a dairy cream to coffee and tea prior to consumption is a common practice” in where? In the U.S. or the other areas?

References:

Line 509: Reference #3 is the same as reference #4.

Line 557: There is a typo in reference “National Cancer Institute),”.

Author Response

Response to Reviewer 1

Introduction

Line 36: There might be a typo in the statement ‘coffee of tea with no added sweeteners’.

Response: Correct, this was a typo and has been corrected in the revised manuscript. The text has been revised to “coffee or tea with no added sweeteners.”

Lines 64-68: This statement seems to be long to be recognized.

Response: This sentence has been split into two sentences and edited in the revised manuscript. The text has been revised to (with revised text colored red):

 “Available data on beverage consumption patterns in the U.S. therefore suggest that patterns including water or LNCSB are associated with beneficial trends in nutrient intakes.  , though However, data are lacking on comparisons between consumers of water or LNCSB and consumers of SSB to allow for a clear understanding of how these water-based beverage patterns differ in overall diet quality and how this understanding may help consumers achieve healthier dietary patterns.” 

Methods

Lines 125-126: It is not clear about the statement “the alternative water-based alternative beverage options (i.e., ZCUB or LNCSB)”.

Response: We have edited the sentence to not have the word “alternative” repeated twice.  This was a typographical error.  Thank you.

Results

Lines 193-205: The results of Figure 1-d (HEI-2015 overall score) and Figure 2-d seem not to be reported.

Response: Figures 1-d and 2-d are not related to nutrients (as the header of this section indicates) but rather diet quality.  Therefore, it is not appropriate to reference those figures in this section.  A reference to Figures 1d and 2d have been added to the Diet Quality sections in the revised manuscript.

Lines 218-220: The statement “mean total sugar and carbohydrate reductions” should be “the differences in mean total sugar and carbohydrate reductions”.

Response: This edit has been made in the revised manuscript.

Lines 246-247: The statement “the relative difference in added sugar scores” is not clear. It could be revised as “ …the relative difference in HEI-2015 component scores for added sugar in adults aged 19-64 years”.

Response: This edit has been made in the revised manuscript.

Line 275: The authors need to explain why the number for “SSB only” is 4,637 in Table 3 and 17,554 in Table 6. In Figure 2, statistically significant declining trends in the intakes of total and added sugars and carbohydrates were detected among “SSB only”, but in Figure 1, there were no such declining trends for “SSB only” consumers. In the discussion section (lines 420-429), the authors offered an explanation for this issue. Could the authors display the percentage distribution for the categories that are overlapped and not overlapped for the “SSB only” group in these two series of analyses.

 Response: As the reviewer indicates, we have discussed the difference in the number of individuals within the SSB only group within the discussion and we believe this is the most appropriate place to discuss this data point.  These are two independent analyses with two independent definitions of beverage consumer classifications.  We have added the number of individuals within each category to Table 2 to further highlight this point.   

Line 279: The subtitle should be “3.2.2. Nutrient Intakes”.

Response:  Thank you; this has been edited in the revised manuscript.

Line 346: There is an unbalanced quotes in this statement “LNCSB only consumers’ added”

Response: The apostrophe on consumers is used to imply it is this group of consumers’ added sugar intake.  It is not intended to be a quotation.

Lines 362-363: “sugar-sweetened 362 beverages” can be abbreviated to be “SSB”.

Response:  Thank you; this has been edited in the revised manuscript.

Lines 374-376: This statement needs to cite references to denote “similar results”.

Response: This statement refers to the LNCSB analysis that is part of the current manuscript.  It is not an outside reference.  No edit was made to the manuscript.

Lines 404-405: “the addition of sugar and/or a dairy cream to coffee and tea prior to consumption is a common practice” in where? In the U.S. or the other areas?

Response:  Yes, in the US.  The reference cited is among US adults.  This has been edited in the revised manuscript. The sentence has been revised to (with revised text in red): “Nevertheless, the addition of sugar and/or a dairy cream to coffee and tea prior to consumption is a common practice in the U.S. [24]”. 

References:

Line 509: Reference #3 is the same as reference #4.

Response: These are two different references – one for Adults and one for Youth. 

Line 557: There is a typo in reference “National Cancer Institute),”.

Response:  Thank you; this has been edited in the revised manuscript.

Reviewer 2 Report

I read with interest the publication by Barraj and colleagues and have some minor suggestions for the authors.

Page 2 line 51: Insert some references to support this sentence “…. higher measures of diet quality in some (reference) though not all studies (reference)

Page 2 line 93-94: switch the order to “non breast feeding females and males”.

Page 3. Overall I found the classification of water beverages clear with the exception of lines 111 to 114. For instance the brewed coffee with sugar and cream was not classified as an SSB due to the presence of cream.  This decision seems a little arbitrary to me given that the composition of cream would contribute little to added sugar or carbohydrate intake for example, and yet, sugar (which was in the coffee) would be contributing to added sugar intake. I would have thought that beverages with sugar would have been classified with SSB. Could the authors explain the rationale further for this classification or provide the reader with the frequency of beverages that were classified as ‘other’ within each of these domains (i.e. the brewed coffee with sugar and cream and the lemon or lime juice)  The classification ‘other beverages’ could be added to Table 1 classification. This would require changing the description of table 1 to “Classification of Beverages”.

A general question: Is zero calorie unsweetened beverage standard terminology? In the classification of beverages could zero-calorie unsweetened beverage be changed to zero calorie beverages?

Page 4: Table 2 the number of observations within each group could be specified in this table. With the description of “Both” could the authors change to SSB+ZCUB? And SSB+LNCSB. Also rather than using the term analysis in Table 2 I think the correct terminology to use here would be Classification 1 and Classification 2.

Page 5: In statistical the methods specify what the p-value was for the Bonferroni correction.

Perhaps one major consideration that is inadequately addressed in the paper is the impact of underreporting, and whether the authors considered exclusion of data based on implausible energy intakes. For example the energy intake reported in the neither group is low in Table 3 for example compared to the other beverage categories.

Page 11 line 334: Edit “….adult “LNCSB only” consumers had on average a 3.9 points higher r HEI score compared to SSB only consumers”.

Suggest removing some of the text if the focused is on the supplementary tables, such as on page 12 line 341 to 350. With the description of added sugar scores closer to two times higher (i.e. lower added sugar) the overall context of this paragraph is consuming without the actual data presented in the manuscript. Do the authors feel the description of the supplementary data enhance the overall take-home message of the paper? There is a lot of results to digest and this paper is long.

Author Response

Response to Reviewer 2

Page 2 line 51: Insert some references to support this sentence “…. higher measures of diet quality in some (reference) though not all studies (reference)

Response: Citations have been added to the revised manuscript. The text has been revised to “… higher measures of diet quality in some [8, 9] though not all [10] studies.”

Page 2 line 93-94: switch the order to “non breast feeding females and males”.

Response: This edit has been made in the revised manuscript.

Page 3. Overall I found the classification of water beverages clear with the exception of lines 111 to 114. For instance the brewed coffee with sugar and cream was not classified as an SSB due to the presence of cream.  This decision seems a little arbitrary to me given that the composition of cream would contribute little to added sugar or carbohydrate intake for example, and yet, sugar (which was in the coffee) would be contributing to added sugar intake. I would have thought that beverages with sugar would have been classified with SSB. Could the authors explain the rationale further for this classification or provide the reader with the frequency of beverages that were classified as ‘other’ within each of these domains (i.e. the brewed coffee with sugar and cream and the lemon or lime juice)  The classification ‘other beverages’ could be added to Table 1 classification. This would require changing the description of table 1 to “Classification of Beverages”.

Response: Our objective and analyses were focused on water-based beverages defined as LNCSB, unsweetened water, unsweetened coffee, unsweetened tea, SSB, or a sweetener (either caloric or a LNCS) with no other additions to the beverage.  For this reason, milks were excluded in the definition of an SSB/LNCSB/ZCUB.  Therefore, if a coffee or tea had a cream or milk addition, these were included in the “other” category. When conducting these types of analyses, there need to be definitions and cut-offs in order to make classifications.  Our decision was to remove these given the dairy component and provide a clear description of what was included and excluded for use in interpretation and understanding of results.   

A general question: Is zero calorie unsweetened beverage standard terminology? In the classification of beverages could zero-calorie unsweetened beverage be changed to zero calorie beverages?

Response: The terminology for these types of beverages is generally not standardized among studies and publications.  We defined each beverage category with an assigned name and acronym with the objective to provide a highly specific definition of each consumer group within beverage category

Page 4: Table 2 the number of observations within each group could be specified in this table. With the description of “Both” could the authors change to SSB+ZCUB? And SSB+LNCSB. Also rather than using the term analysis in Table 2 I think the correct terminology to use here would be Classification 1 and Classification 2.

Response: We have edited the table based on your comment and have added the number of consumers within each group as shown below (with revised text colored red):

Table 2. Definition of water-based beverage consumer groups.

Classification   for Analysis

Water-based Beverage   Consumer Group (N=number of consumers)

Description

Analysis #1: ZCUB

SSB

ZCUB

"SSB only" (N=4,637)

Yes

No

"Both" (N=14,414)

Yes

Yes

"ZCUB only" (N=12,452)

No

Yes

"Neither" (N=1,456)

No

No

Analysis #2: LNCSB

SSB

LNCSB

"SSB only" (N=17,554)

Yes

No

"Both" (N=1,497)

Yes

Yes

"LNCSB only"   (N=3,215)

No

Yes

"Neither" (N=10,693)

No

No

Page 5: In statistical the methods specify what the p-value was for the Bonferroni correction.

Response: The p-value cut-off was 0.05; the p-values provided in the statistical output were the p-values adjusted for multiple comparisons and are noted in the result tables, where appropriate. 

Perhaps one major consideration that is inadequately addressed in the paper is the impact of underreporting, and whether the authors considered exclusion of data based on implausible energy intakes. For example the energy intake reported in the neither group is low in Table 3 for example compared to the other beverage categories.

Response:  Similar to other analyses conducted using NHANES, we did not exclude participants based on energy intakes.  We did restrict the study population to those individuals with valid day 1 dietary records as determined by NCHS.  The lower mean energy intake among “Neither” consumers as well as the ZCUB only consumers (who actually have the lowest mean energy intakes in this analysis), is correlated, in part, with their lower added sugar intake and reflected by a higher diet quality.  The issue with under-reporting is also commented on in the discussion.

Page 11 line 334: Edit “….adult “LNCSB only” consumers had on average a 3.9 points higher r HEI score compared to SSB only consumers”.

Response:  This edit has been made to the revised manuscript.

Suggest removing some of the text if the focused is on the supplementary tables, such as on page 12 line 341 to 350. With the description of added sugar scores closer to two times higher (i.e. lower added sugar) the overall context of this paragraph is consuming without the actual data presented in the manuscript. Do the authors feel the description of the supplementary data enhance the overall take-home message of the paper? There is a lot of results to digest and this paper is long.

Response: The authors do feel this description of the individual components of the HEI are important to the overall message and would prefer to keep the text in with a reference to the data in the supplement.